# Numerical Modelling Analysis for Carrier Concentration Level Optimization of CdTe Heterojunction Thin Film–Based Solar Cell with Different Non–Toxic Metal Chalcogenide Buffer Layers Replacements: Using SCAPS–1D Software

**Samer H. Zyoud** [1,2,3,]*, **Ahed H. Zyoud** [4], **Naser M. Ahmed** [3] and **Atef F. I. Abdelkader** [1,2]

1 Department of Mathematics and Science, Ajman University, Ajman P.O. Box 346, United Arab Emirates; a.abdelkader@ajman.ac.ae
2 Nonlinear Dynamics Research Center (NDRC), Ajman University, Ajman P.O. Box 346, United Arab Emirates
3 School of Physics, Universiti Sains Malaysia, Penang 11800, Malaysia; naser@usm.my
4 Department of Chemistry, An-Najah National University, Nablus, Palestine; ahedzyoud@najah.edu
* Correspondence: s.zyoud@ajman.ac.ae

**Abstract:** Cadmium telluride (CdTe), a metallic dichalcogenide material, was utilized as an absorber layer for thin film–based solar cells with appropriate configurations and the SCAPS–1D structures program was used to evaluate the results. In both known and developing thin film photovoltaic systems, a CdS thin–film buffer layer is frequently employed as a traditional *n*–type heterojunction partner. In this study, numerical simulation was used to determine a suitable non–toxic material for the buffer layer that can be used instead of CdS, among various types of buffer layers (ZnSe, ZnO, ZnS and In$_2$S$_3$) and carrier concentrations for the absorber layer ($N_A$) and buffer layer ($N_D$) were varied to determine the optimal simulation parameters. Carrier concentrations ($N_A$ from $2 \times 10^{12}$ cm$^{-3}$ to $2 \times 10^{17}$ cm$^{-3}$ and $N_D$ from $1 \times 10^{16}$ cm$^{-3}$ to $1 \times 10^{22}$ cm$^{-3}$) differed. The results showed that the use of CdS as a buffer–layer–based CdTe absorber layer for solar cell had the highest efficiency (%) of 17.43%. Furthermore, high conversion efficiencies of 17.42% and 16.27% were for the ZnSe and ZnO-based buffer layers, respectively. As a result, ZnO and ZnSe are potential candidates for replacing the CdS buffer layer in thin–film solar cells. Here, the absorber (CdTe) and buffer (ZnSe) layers were chosen to improve the efficiency by finding the optimal density of the carrier concentration (acceptor and donor). The simulation findings above provide helpful recommendations for fabricating high–efficiency metal oxide–based solar cells in the lab.

**Keywords:** absorber layer and buffer layer; CdTe; ZnSe; conversion efficiency; SCAPS–1D; solar cell

## 1. Introduction

The challenge of global warming has prompted the further study of solar and other renewable energy sources. Solar cells are a fundamental component of solar energy. Different materials are used to create solar cells, with silicon being the most commercially feasible and prevalent. The majority of the alternative materials were developed with the goal of producing low–cost, high–efficiency and long–lasting solar cells. Although the efficiency is still modest, nanostructured metal oxide solar cells have moved a step farther in delivering clean, cheap and sustainable solar cells [1]. Solar energy conversion to useable power using a solid–state *p–n* junction based photovoltaic (PV) device offers enormous promise in the efforts to reduce our current reliance on fossil fuels and, as a result, to reduce harmful greenhouse gas emissions [2,3].

Due to its unique properties, cadmium telluride (CdTe) thin film is widely employed in a variety of optical and electrical applications. CdTe thin–film cells are gaining popularity because of their abundance, excellent efficiency, long–term stability and low cost of manufacture [4] and they can be used in a variety of devices such as nanodevices, sensors

and solar cells [5]. CdTe is classified as an II–VI transition metallic dichalcogenide and has a high absorption coefficient ($>10^5$ cm$^{-1}$) that is greater than other known semiconductor materials with a narrow band gap (Eg~1.5 eV) [6,7]. This band gap value is suitable for the visible solar light spectrum [8–10].

Thin–film solar cell applications make extensive use of metal chalcogenide semiconductors such as cadmium telluride (CdTe). This is due to its inexpensive cost, abundance on the planet and reasonable conversion efficiency. Because of its excellent performance, outdoor long–term stability, strong optical absorption, low cost, adjustable bandgap and unique optoelectronic features, CdTe was chosen as the absorber layer in this work. Various attempts have been made on CdTe to improve its stability and conversion efficiency [11]. In the form of *p*–type semiconductors, CdTe is a potential absorbing material for thin–film PV technology [12]. Despite the widespread usage of CdTe thin films, its primary form has a low conversion efficiency in PEC procedures. When electrodeposited on Ni substrates, CdTe thin films have poor conversion efficiency, depending on the redox couplings and the type of conduction utilized [13]. PEC performance was also poor when CdTe thin films were formed on FTO and ITO substrates [14,15]. When CdTe films were deposited by spray pyrolysis [16], the conversion efficiency was 3.4%, whereas chemical bath–formed films that had been treated with CdCl$_2$ had a conversion efficiency of 2.5% [17–19]. CdTe thin films have been reported to have a conversion efficiency of 17.5% or more under certain circumstances [9]. To increase low PEC performance, CdTe thin films are frequently combined with other systems, such as CdS films [14,20]. Cadmium sulfide (CdS) is a well–known II–VI compound semiconductor with excellent transparency, a straight band gap transition (Eg~2.4 eV), strong electron affinity (~4.2 eV) and *n*-type conductivity [21,22]. CdS also enhances the interface fit of lattice heterojunctions, increases the surplus carrier lifetime and optimizes the band alignment of the devices in which it is utilized [23]. The optical, electrical and structural properties of CdS films are useful in a wide variety of scientific, technical and commercial applications involving optoelectronic devices, particularly solar cells [24]. CdS is a promising option for use as a buffer layer in CdTe thin film–based solar cells due to its properties of low surface recombination and little absorption loss. On the other hand, CdS can be hazardous to the environment and human health due to its high toxicity. Different materials with a larger band gap as well as non–toxic compounds such as ZnS (O, OH) and ZnS have been studied as suitable buffer layers for thin–film solar cells [25–27]. However, because of the complex reaction mechanism and light soaking effects of these buffer layers, cell durability and repeatability may be compromised [28].

CdS/CdTe thin films produced on ITO substrates have been shown to have a conversion efficiency of 3.5% and when silver (Ag) was coated on the films, the conversion efficiency increased to 9.82% [29]. In multi–junction CdTe/CdS combinations, conversion efficiencies of up to 13% have also been recorded [30]. Multi–junction CdS/CdTe/ZnTe/ZnTe:Cu cells have a high conversion efficiency of 13.38% [31,32]. The efficiency of the CdS/CdTe:Cu/CNT structure has been reported to be up to 14.1% [33]. The buffer layer connects the absorber and window layers and it is important for a variety of reasons, including providing structural stability for the thin film and preventing static electricity in the absorber layer [34,35].

In heterojunction thin–film solar cells, the buffer layer generally serves as a focus point. The photons that reach the absorption layer through the reach–in layer travel via the buffer layer. As a result, the number of photons that is lost due to absorption in the buffer layer should be kept to a minimum. As a result, in the buffer layer, electrical resistance and minimal surface recombination are required. In order to provide the buffer layer between the absorber layer and the transparent window layer, it is necessary to provide thin–film solar cell stability. As a result, the buffer layer must have a large energy gap. This permits the majority of visible light to pass through to the absorption layer. On the other hand, for the depletion layer to overlap, the bandgap margins of the buffer and the absorption layer should be roughly compatible. In heterojunction thin–film solar cells, metal chalcogenides such as CdS, CdSe, ZnS, ZnSe and In$_2$S$_3$ are ideal for the role of a buffer layer. CdS, CdSe and CdTe are the most popular metal chalcogenide compounds that are used in thin–film

solar cells with heterojunctions. These substances are harmful to the environment. Green and less dangerous chemicals (such as ZnS, ZnSe, ZnO, $Zn_{1-x}Mg_xO$ and $In_2S_3$) should be studied and assessed as a substitute for the traditional hazardous semiconductors that are often used in heterojunction thin–film solar cells [36]. Numerical simulations may be used to investigate the influence of various materials on the final properties of solar cells. The results of such numerical research and analyses can be utilized to improve device performance [35,37–39]. The optimum and best structure of thin film–based solar cells is determined by numerical modeling. There is currently a scarcity of thin–film solar cell simulation research. As a result, we have narrowed the scope of our numerical simulation in this work by utilizing SCAPS–1D software to investigate the material that is needed for the buffer layer and for substituting the CdS with another material. A different buffer layer's effect on cell performance was investigated. Different buffer layer materials (CdS, ZnO, ZnSe, $In_2S_3$, ZnS) have been shown to exhibit *J–V* characteristics ($V_{oc}$, $J_{sc}$, *FF* and $\eta$) under standard illumination AM1.5G, 100 mW/cm$^2$, 300 K (Table 1). The primary goal of this research is to replace CdS with a different buffer material. Furthermore, the concentration densities of carriers (acceptor and donor) have been considered in this study.

**Table 1.** The working points and illumination.

| Working Points | Value | Spectrum | AM1.5G Spectrum |
|---|---|---|---|
| Temperature (K) | 300 | Wavelength range (nm) | 200–4000 |
| Bias voltage (V) | 0.00 | Transmission (%) | 100 |
| Frequency (Hz) | $1 \times 10^6$ | Ideal light current (mA/cm$^2$) | 20 |
| Series resistance ($\Omega$ cm$^2$) | 0 | Transmission of attenuation filter (%) | 100 |
| Shunt resistance ($\Omega$ cm$^2$) | $1 \times 10^{30}$ | Ideal light current cell (mA/cm$^2$) | 0 |

## 2. Numerical Modeling and Material Parameters

SCAPS–1D was created at ELIS, University of Ghent and it may be used for free in photovoltaic research investigations [40,41]. The SCAPS–1D structure program is frequently used to model the electrical and optical characteristics of AC and DC heterojunctions. It is primarily designed for CIGS and CdTe solar cells. The main goal of SCAPS–1D is to use an existing database to explore the properties of thin film–based solar cells with various buffer layers. SCAPS–1D simulation solutions may be used to examine outputs such as voltage and currents on illumination and dark characteristics. This simulation may also generate a temperature–based analysis. SCAPS–1D simulation may also provide important information such as recombination profiles, the current density of individual carriers as a positional function and electrical physical distribution. The main goal is to replicate solar cells in order to achieve high efficiency before beginning actual experimental manufacturing with various parameters. SCAPS–1D simulation may be used to investigate the impact of various parameters on ($V_{OC}$, $J_{SC}$, *FF*, $\eta$) and operating temperature.

### 2.1. Numerical Modeling

SCAPS–1D can solve Poisson's equation for holes and electrons (Equation (1)) [42]:

$$\frac{d^2\Psi}{dx^2} = \frac{e}{\epsilon_o\epsilon_r}\left[P(x) - n(x) + N_D - N_A + \rho_P - \rho_n\right] \tag{1}$$

where $\Psi$ is the electrostatic potential, *e* is the elementary charge, $\epsilon_r$ is the relative permittivity and $\varepsilon_o$ is the vacuum permittivity, *p* is hole concentration, *n* is electron concentration, $N_D$, $N_A$ are donor and acceptor charge concentrations, respectively and $\rho_p$ and $\rho_n$ are holes and electrons distribution, respectively.

Additionally, it can also solve the continuity equation (Equation (2)) [43]:

$$\frac{d^2\Psi}{dx^2} = \frac{e}{\epsilon_o\epsilon_r}\left[P(x) - n(x) + N_D - N_A + \rho_P - \rho_n\right] \tag{2}$$

where $J_p$ and $J_n$ are the hole and electron current densities, respectively and $R$ and $G$ are recombination rates, respectively.

Carrier transport occurs by drift and diffusion according to Equations (3) and (4), respectively:

$$J_n = D_n \frac{dn}{dx} + \mu_n n \frac{d\varphi}{dx} \tag{3}$$

$$J_p = D_p \frac{dp}{dx} + \mu_p p \frac{d\varphi}{dx} \tag{4}$$

where $\varphi$ is the potential difference and $D_n$ and $D_p$ are the electron and hole diffusion constant, respectively. $\mu_n$ and $\mu_p$ are the electron and hole mobility and $n$ and $p$ are the electron and hole carrier concentration.

### 2.2. The Suggested Thin-Film Solar Cell Device Structure

Figure 1 shows the thin film's structure, which includes a *p*–type absorber (CdTe) layer on a molybdenum (Mo) coated back glass substrate, an *n*–type buffer layer (CdS, In$_2$S$_3$, ZnS, ZnO, ZnSe) and a SnO$_2$ window layer.

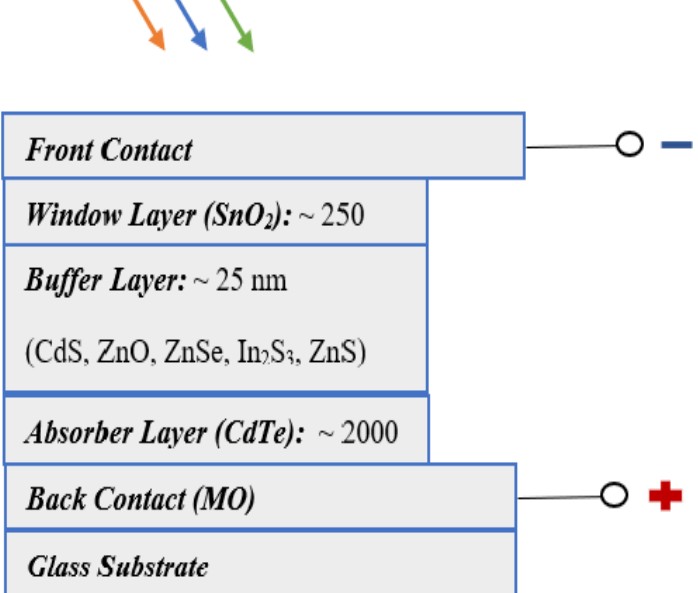

**Figure 1.** Schematic diagram of thin film.

### 2.3. Numerical Material Parameters

The starting conditions (bias voltage, operating temperature, lighting and so on) should be established, as stated in Table 1, at the start of the simulation. Table 2 shows the reflection and transmission of the front and back contacts, respectively. Each layer's material characteristics should be entered into a software application. Table 3 shows the material parameter characteristics for the thin film layers and lists the physical parameters utilized in the Mo/absorber/buffer/window solar cell simulation [44–61].

**Table 2.** Electrical parameters properties of back and front contact used for the metal oxide SCAPS–1D simulation.

| Electrical Properties | | Back Contact | Front Contact |
|---|---|---|---|
| Thermionic emission surface recombination velocity (cm/s) | Electron | $1 \times 10^7$ | $1 \times 10^7$ |
| | Holes | $1 \times 10^7$ | $1 \times 10^7$ |
| Metal work function (eV) | | 5 | 4.1 |
| Majority carrier barrier height (eV) | Relative to $E_F$ | 0.4 | 0.1 |
| | Relative to $E_V$ or $E_C$ | −0.1227 | 0.0199 |
| Allow contact tunneling | Effective mass of electron | 1 | 1 |
| | Effective mass of holes | 1 | 1 |
| Optical Properties | Filter mode | Reflection | Transmission |
| | Filter value | 0.8 | 0.95 |
| | Complement of filter value | 0.2 | 0.05 |

**Table 3.** The electrical parameters for the thin–film solar cell at 300 K.

| Electrical Parameter | $p$–CdTe | $n$–CdS | $n$–ZnS | $n$–ZnSe | $n$–ZnO | $n$–In$_2$S$_3$ | SnO$_2$ |
|---|---|---|---|---|---|---|---|
| Thickness (μm) | 2 | 0.025 | 0.025 | 0.025 | 0.025 | 0.025 | 0.25 |
| Band gap (eV) | 1.5 | 2.4 | 3.5 | 2.9 | 3.3 | 2.8 | 3.6 |
| Electron affinity (eV) | 3.9 | 4.5 | 4.5 | 4.09 | 4.45 | 4.7 | 4 |
| Dielectric permittivity (relative) | 9.4 | 10 | 10 | 10 | 9 | 13.5 | 9 |
| CB effective density of states (cm$^{-3}$) | $8 \times 10^{17}$ | $1.5 \times 10^{18}$ | $1.5 \times 10^{18}$ | $1.5 \times 10^{18}$ | $2.2 \times 10^{18}$ | $1.8 \times 10^{19}$ | $2.2 \times 10^{18}$ |
| VB effective density of states (cm$^{-3}$) | $1.8 \times 10^{19}$ | $1.8 \times 10^{18}$ | $1.8 \times 10^{18}$ | $1.8 \times 10^{18}$ | $1.8 \times 10^{19}$ | $4 \times 10^{18}$ | $1.8 \times 10^{18}$ |
| Electron thermal velocity (cm/s) | $1 \times 10^7$ | $1 \times 10^7$ | $1 \times 10^7$ | $1 \times 10^7$ | $1 \times 10^7$ | $1 \times 10^7$ | $1 \times 10^7$ |
| Hole thermal velocity (cm/s) | $1 \times 10^7$ | $1 \times 10^7$ | $1 \times 10^7$ | $1 \times 10^7$ | $1 \times 10^7$ | $1 \times 10^7$ | $1 \times 10^7$ |
| Electron mobility (cm$^2$/V s) | 300 | 50 | 50 | 50 | 100 | 400 | 100 |
| Hole mobility (cm$^2$/V s) | 40 | 20 | 20 | 20 | 25 | 210 | 25 |
| Shallow uniform donor density $N_D$ (cm$^{-3}$) | 0 | $1 \times 10^{22}$ | $1 \times 10^{22}$ | $1 \times 10^{22}$ | $1 \times 10^{22}$ | $1 \times 10^{22}$ | $1 \times 10^{22}$ |
| Shallow uniform acceptor density $N_A$ (cm$^{-3}$) | $2 \times 10^{15}$ | 0 | 0 | 0 | 0 | 0 | 0 |

## 3. Results and Discussion

### 3.1. Effect of Different Buffer Layer on Thin Film-Based Solar Cell

Cadmium (Cd) is poisonous and CdS is classified as a carcinogen, both of which are harmful to the environment and humans. Other potential buffer layers, such as ZnO, In$_2$S$_3$, ZnSe and ZnS, have been explored as a result. The optimal photovoltaic parameters ($V_{OC}$, $J_{SC}$, *FF* and $\eta$ %) of a CdTe thin film with various buffer layers are shown in Table 4 and Figure 2. It should be highlighted that CdS performs the best as a buffer layer, obtaining an efficiency of 17.43%. The results also reveal that buffer layers made of ZnSe and ZnO have excellent efficiency, at 17.42% and 16.27%, respectively. While buffer layers based on ZnS and In$_2$S$_3$ had a lower efficiency of 15.88% and 14.23%, respectively. As a result, ZnO and ZnSe have been proposed as replacements for CdS as a buffer layer in thin films [62].

Figure 3a depicts the *J–V* characteristics for various buffer layers. It is worth noting that when the efficiency is high, the curve shifts to the right.

**Table 4.** Effectiveness of the buffer material (donor) on *J–V* characteristics.

| Buffer Layer | $V_{OC}$ (V) | $J_{SC}$ (mA/cm$^2$) | *FF* | Efficiency (%) |
|---|---|---|---|---|
| CdS | 0.9113 | 23.4497335 | 81.41 | 17.43 |
| ZnSe | 0.9112 | 23.484037 | 82.38 | 17.42 |
| ZnO | 0.9142 | 23.303926 | 76.37 | 16.27 |
| ZnS | 0.9121 | 23.260166 | 74.84 | 15.88 |
| In$_2$S$_3$ | 0.9198 | 23.153579 | 66.81 | 14.23 |

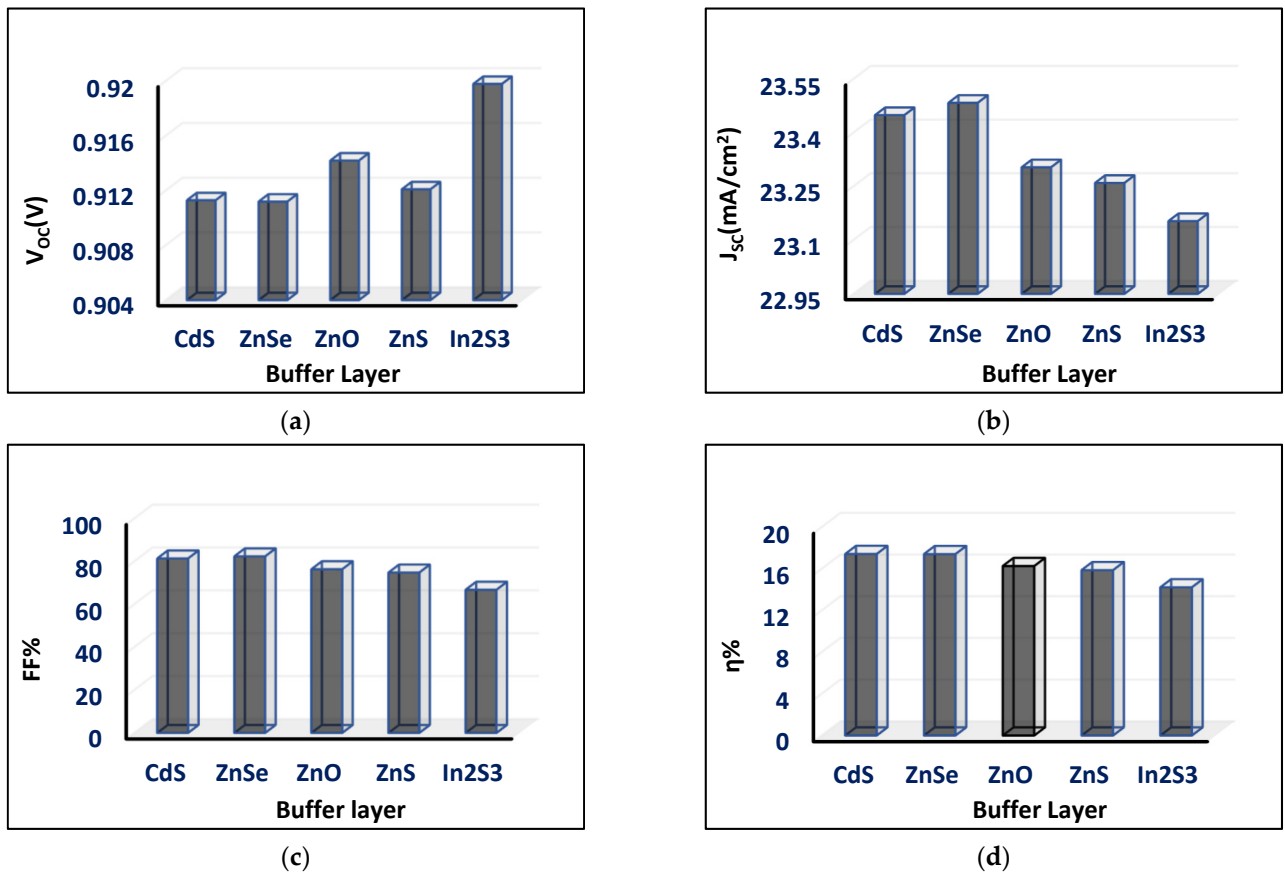

**Figure 2.** Effectiveness of the buffer material (donor) layer on photovoltaic parameters (**a**) $V_{OC}$ (**b**) $J_{SC}$ (**c**) $FF$ (**d**) $\eta$.

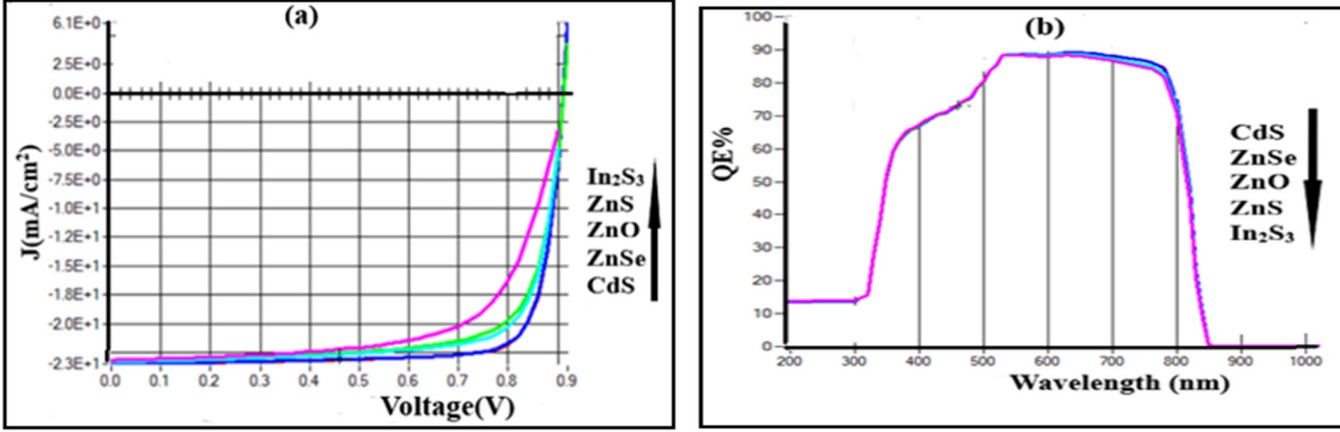

**Figure 3.** (**a**) $J$–$V$ current curves for the cell; (**b**) spectral response of solar cells and CdTe layer (acceptor characteristics) with different buffer layer at T = 300 K.

The following equation, Equation (5), can be used to calculate the spectrum response using the external quantum efficiency:

$$EQE(\lambda) = \frac{\frac{I(\lambda)}{q}}{\varphi_p(\lambda)} \tag{5}$$

where $q$ represents the fundamental electrical charge, $I(\lambda)$ represents the photogenerated current and $\varphi_p(\lambda)$ represent the photon flux. On the light spectrum, Figure 4 depicts the external quantum efficiency $QE$ (%) for various buffer layers. The results reveal that when

the buffer layer is CdS, the efficiency is at its peak [63]. The impact of the different buffers on the light spectrum might be seen in Figure 3b.

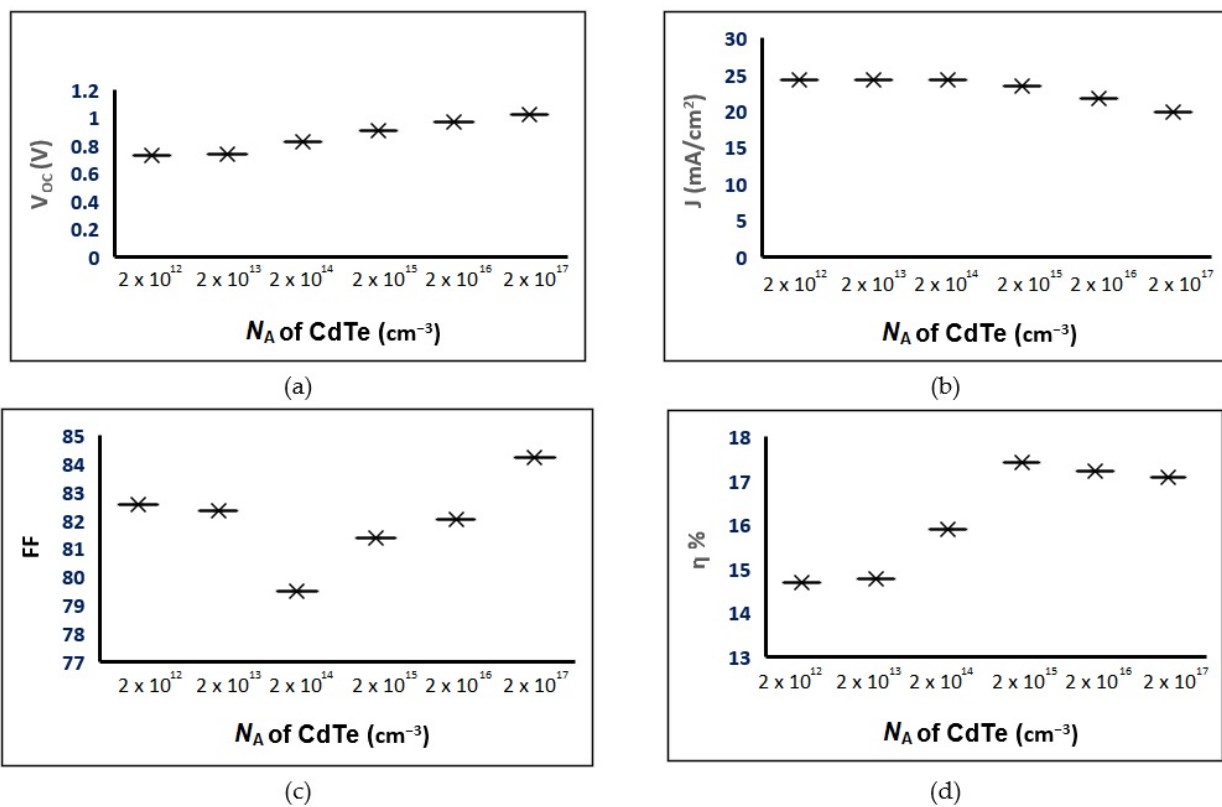

**Figure 4.** The simulated electrical performance parameters as a function of the acceptor charge carrier concentration ($N_A$): (**a**) $V_{OC}$, (**b**) $J_{SC}$, (**c**) *FF*, (**d**) $\eta$%.

### 3.2. Modelling and Optimization of CdTe Absorber Layer Doping Level

The absorber (CdTe) acceptor carrier concentration ($N_A$) changes in the ranges of $2 \times 10^{12}$ cm$^{-3}$ to $2 \times 10^{17}$cm$^{-3}$, as shown in Table 5. The main goal of this study is to obtain a carrier concentration ($N_A$) of the CdTe absorber layer without losses in cell performance. As consequence, low cost can be accomplished by reducing the amount of expensive materials that is used.

**Table 5.** Effectiveness of the acceptor carrier concentration ($N_A$) on the electrical cell performance parameters at T = 300 K.

| $N_A$ (cm$^{-3}$) | $V_{OC}$ (V) | $J_{SC}$ (mA/cm$^2$) | *FF*% | $\eta$ (%) |
|---|---|---|---|---|
| $2 \times 10^{12}$ | 0.7333 | 24.249669 | 82.57 | 14.68 |
| $2 \times 10^{13}$ | 0.7398 | 24.248777 | 82.33 | 14.77 |
| $2 \times 10^{14}$ | 0.8263 | 24.223452 | 79.49 | 15.91 |
| $2 \times 10^{15}$ | 0.9113 | 23.484037 | 81.38 | 17.42 |
| $2 \times 10^{16}$ | 0.9662 | 21.748835 | 82.02 | 17.23 |
| $2 \times 10^{17}$ | 1.0247 | 19.810091 | 84.21 | 17.09 |
| $2 \times 10^{12}$ | 0.7333 | 24.249669 | 82.57 | 14.68 |

The electrical parameter performance with an acceptor (hole) carrier charge concentration ($N_A$) when the CdTe absorber has a 2000 nm thickness of is shown in Figure 4a–d. Figure 4a depicts a linear rise in an open–circuit voltage ($V_{OC}$) with ($N_A > 2 \times 10^{14}$ cm$^{-3}$). Figure 4b depicts a linear reduction in the short–circuit current density ($J_{SC}$) with ($N_A > 2 \times 10^{14}$ cm$^{-3}$); this can be ascribed to an increase in free carrier charge recom-

bination inside the bulk [64]. On the other hand, the fill factor (*FF*), as shown in Figure 4c, increases linearly with ($N_A > 2 \times 10^{14}$ cm$^{-3}$). Figure 4d also demonstrates that a low hole doping level ($N_A < 2 \times 10^{15}$ cm$^{-3}$) leads to a significant reduction in the device conversion efficiency, with values of less than 3%. When the hole concentration of the absorber layer increases, however, minor cell efficiency changes are found, as shown by Equations (6)–(9):

$$J_{SC} = q \sum T(\lambda) \frac{\varnothing_i(\lambda_i)}{h\nu_i} \eta(\lambda_i) \Delta\lambda_i \tag{6}$$

where $q$ denotes the elementary charge, $\varnothing_i$ denotes the spectral power density, $T(\lambda)$ denotes the optical transmission and $\Delta\lambda_i$ denotes the distance between two adjacent wavelength values.

$$V_{OC} = \frac{nkT}{q} \ln\left(\frac{J_{SC}}{J_O} + 1\right) \tag{7}$$

$$FF\% = \frac{V_{OC} - \ln(V_{OC} + 0.72)}{V_{OC} + 1} \tag{8}$$

$$\eta\% = \frac{V_{OC} \times J_{SC} \times FF\%}{P_{in}} \tag{9}$$

The improved efficiency (Figure 4d) in the simulated findings is explained by the combined impact of current density $J_{SC}$ saturation (Figure 4b) as well as the rapid increase of $V_{OC}$ and *FF* (Figure 4a,c) with the acceptor carrier charge concentration ($N_A$). As a result, ($N_A \sim 2 \times 10^{15}$ cm$^{-3}$) provides the best performance for the CdTe thin film.

The effect of the changes in the CdTe acceptor charge carrier concentration ($N_A$) on solar cell fundamental characteristics was thoroughly explored. The thin film's spectral response to the CdTe acceptor carrier charge concentration ($N_A$) is shown in Figure 5a. The simulated findings show that when the acceptor concentration increases from $2 \times 10^{12}$ cm$^{-3}$ to $2 \times 10^{19}$ cm$^{-3}$, the quantum efficiency (*QE*%) decreases. The enhanced gathering of photons at longer wavelengths can be ascribed to this. The production of additional pairs of electron holes in the thin–film solar cell results from the absorption of longer wavelength photons, resulting in an increase in $J_{SC}$ at low acceptor charge carrier concentrations ($N_A$) (Figure 5b). The *J–V* curves show that the $V_{OC}$ increases as the acceptor charge carrier concentration ($N_A$) of the CdTe layer increases ($N_A > 2 \times 10^{14}$ cm$^{-3}$). This rise shows that the open–circuit voltage ($V_{OC}$) of the CdTe layer is substantially influenced by the acceptor charge carrier concentration ($N_A$). The generated electric field in the depletion region is altered when the acceptor (hole) carrier charge concentration ($N_A$) of the CdTe layer is high [65]. As a result, the free charge carrier recombination decreased, increasing the $V_{OC}$. While lowering the CdTe acceptor carrier charge concentration below $2 \times 10^{15}$ cm$^{-3}$ results in increased optical losses, which might be due to surface recombination at the back contact [66].

The following Equations (8) and (9) explain the *p–n* junction model:

$$I_O = Aqn_i^2\left(\frac{D_e}{L_e N_A} + \frac{D_h}{L_h N_D}\right) \tag{10}$$

$$V_{OC} = \frac{KT}{qln(I_L/I_O)} \tag{11}$$

$I_O$ denotes the saturation current, $n_i$ is the intrinsic concentration, $A$ is the diode quality factor, $q$ is the elementary charge, $T$ is the temperature, $k$ is the Boltzmann constant, $I_L$ is the light–generated current, $L$ and $D$ are the diffusion length and coefficient and $N_D$ and $N_A$ are the donor and acceptor charge concentrations. The letters $h$ and $e$ stand for holes and electrons, respectively. As the acceptor carrier concentration $N_A$ rises, the saturation current $I_O$ decreases, resulting in an increase in $V_{OC}$ and a drop in $J_{SC}$. The reason for this is that when the acceptor carrier concentration is high, the recombination

process increases and reduces the probability of electron–hole production pairs, lowering the *QE* (%) of long wavelength photons. Long–wavelength photons will be absorbed profoundly in the absorber (CdTe) layer [67].

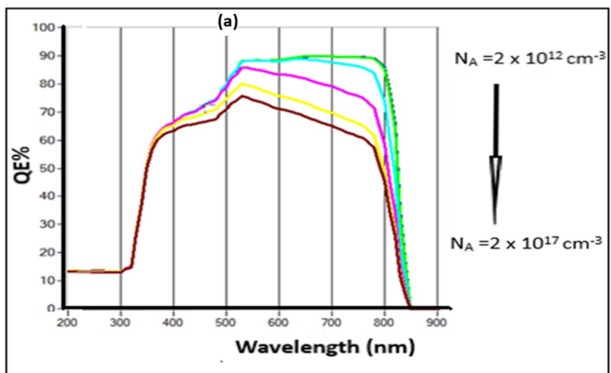 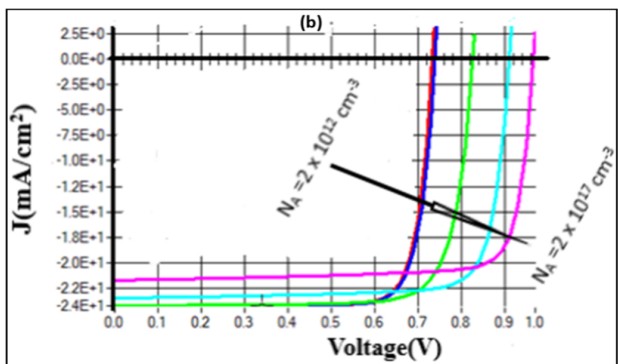

**Figure 5.** (**a**) Spectral response of the enhanced quantum efficiency (*QE*) at longer wavelength and (**b**) *J–V* current curves for the cell with the increase of acceptor carrier concentration ($N_A$).

### 3.3. Modelling and Optimization of ZnSe Buffer Layer Doping Level

The major goal of this section is to decrease the buffer layer's losses (both optical and electrical). Following that, the carrier charge concentration level of the ZnSe layer was adjusted from $1 \times 10^{16}$ cm$^{-3}$ to $1 \times 10^{22}$ cm$^{-3}$. The effect of the ZnSe buffer on the thin–film performance characteristics is shown in Table 6 and Figure 6. With $\left( N_D > 1 \times 10^{18} \text{ cm}^{-3} \right)$, the simulated results show that there is a small amount of modification that can be seen in the cell performance characteristics. The effectiveness of the thin film improved by 3% when the donor concentration increased to $\left( N_D = 1 \times 10^{22} \text{ cm}^{-3} \right)$. It is better to have a high doping level in thin film to retain its exceptional overall performance [68]. The maximum conversion efficiency is 17.42% when the donor carrier charge concentration $(N_D {\sim} 1 \times 10^{22}$ cm$^{-3})$ is used.

Figure 6a–d illustrate how a high donor concentration in the buffer layer improves cell performance. This is due to the apparent requirement for a minimum buffer layer thickness to compensate for the dislocation effect that is caused by the grid mismatches between the ZnSe and CdTe layers. Although the $J_{SC}$, *FF* and $\eta$ parameters all rise (Figure 6b–d), the $V_{OC}$ drops (Figure 6a). The explanation for this may be ascribed to photon loss on a large buffer layer, as seen in Figure 7a. As the concentration of the buffer layer ($N_D$) decreases, more incident photons that are generated by the ZnSe layer are absorbed, reducing the number of photons that the absorber (CdTe) layer can absorb. As illustrated in Figure 7b, absorbed photons generate fewer electron–hole pairs, resulting in a lower *QE* (%). As the donor carrier charge concentrations increase, so does the *QE*. In the simulation, it is better to have a high buffer layer donor concentration ($N_D > 1 \times 10^{18}$ cm$^{-3}$) for thin films.

**Table 6.** Effective of the donor charge carrier concentration ($N_D$) on the electrical cell performance parameters, at T = 300 K.

| $N_D$ (cm$^{-3}$) | $V_{OC}$ (V) | $J_{SC}$ (mA/cm$^2$) | *FF* | $\eta$ (%) |
|---|---|---|---|---|
| $1 \times 10^{16}$ | 0.9178 | 23.275532 | 67.70 | 14.46 |
| $1 \times 10^{17}$ | 0.918 | 23.25272 | 68.22 | 14.57 |
| $1 \times 10^{18}$ | 0.9103 | 23.33507 | 78.62 | 16.70 |
| $1 \times 10^{19}$ | 0.9111 | 23.419193 | 80.54 | 17.19 |
| $1 \times 10^{20}$ | 0.9112 | 23.453013 | 81.07 | 17.33 |
| $1 \times 10^{21}$ | 0.9113 | 23.474864 | 81.29 | 17.39 |
| $1 \times 10^{22}$ | 0.9113 | 23.484037 | 81.38 | 17.42 |

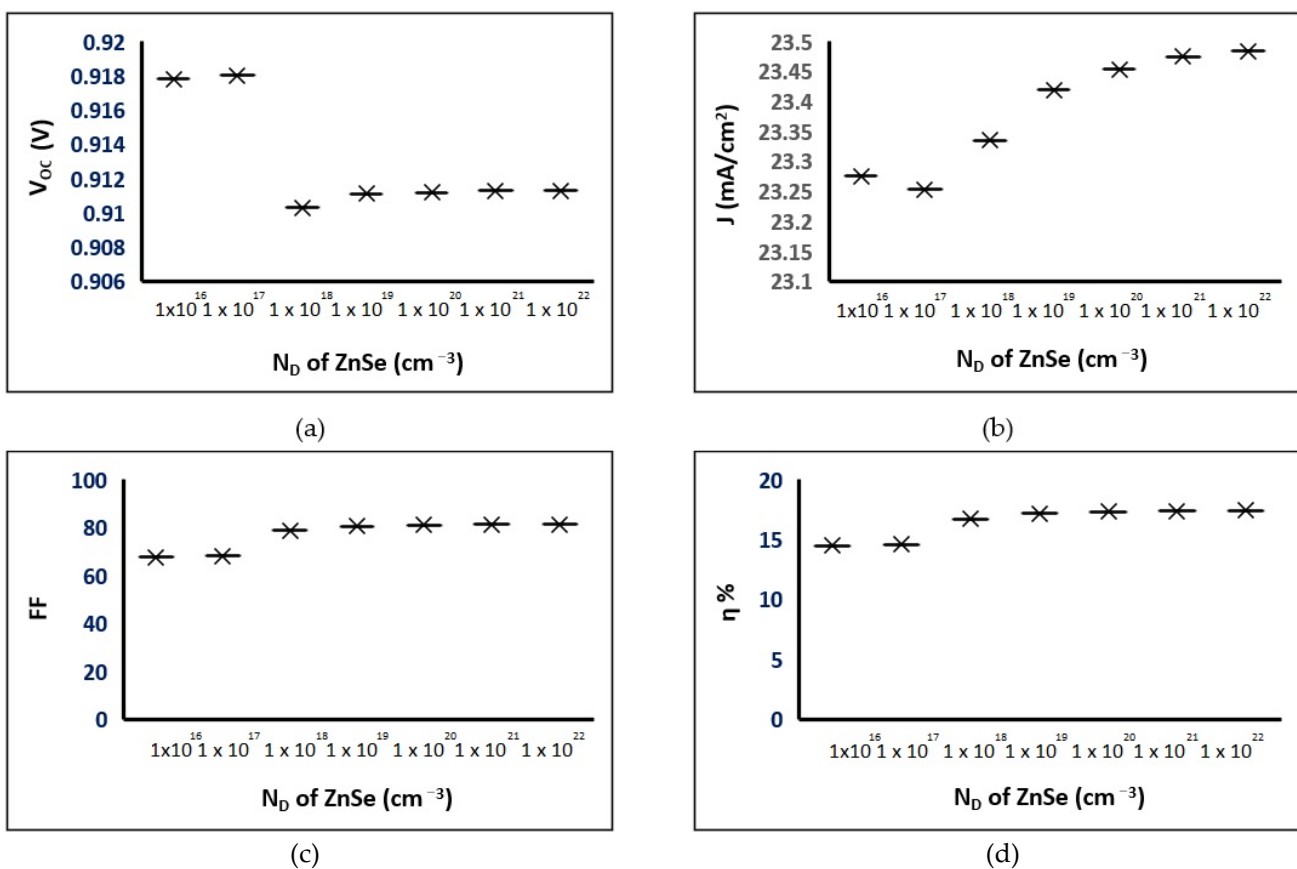

**Figure 6.** The simulated electrical performance parameters as a function of donor charge carrier concentration ($N_D$): (**a**) $V_{OC}$, (**b**) $J_{SC}$, (**c**) *FF*, (**d**) $\eta$.

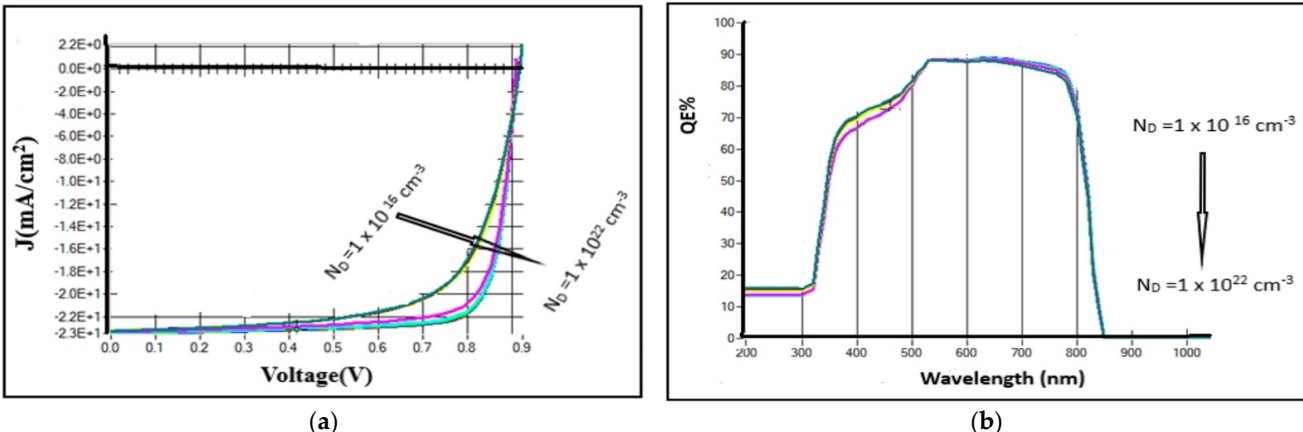

**Figure 7.** (**a**) *J–V* as a function of ZnSe donner carrier concentration ($N_D$). (**b**) Spectral response of the enhanced quantum efficiency (*QE*) at longer wavelength with the increase in the donner carrier concentration ($N_D$).

### 3.4. Optimization of the Mo/CdTe/ZnSe/SnO$_2$ Thin Film-Based Solar Cell

Based on the simulation results described above in the specified parameter range, the optimum PV characteristics can be achieved with an efficiency of 17.42% (with $V_{OC}$ = 0.9113 V, $J_{SC}$ = 23.484037 mA/cm$^2$ and *FF* = 81.38), when the thickness and acceptor concentration of the CdTe are 2000 nm and $2 \times 10^{15}$ cm$^{-3}$, respectively, the thickness and donor concentration of the ZnSe are 25 nm and $1 \times 10^{22}$ cm$^{-3}$ and the thickness and donor concentration of the SnO$_2$ are 250 nm and $1 \times 10^{22}$ cm$^{-3}$, respectively. Other electrical and optical parameters of the thin film are unchanged, as shown in Table 1.

### 3.4.1. Band Diagram

One of the most notable factors impacting thin–film performance and current transmission across heterojunctions is band alignment shown in Figure 8. CdTe is used as the absorber layer (0 to 2 μm), where ZnSe is the buffer layer (2 μm to 2.025 μm) and SnO$_2$ is the window layer (from 2.025 μm to 2.275 μm). When the absorber layer's conduction band is higher than the buffer layer's conduction band, the result is a "cliff" type band alignment [69]. As seen in Figure 8, this is the situation with CdTe thin films. It can be seen that the absorber, buffer and window layers have acceptable band alignment. Four recombination regions can be seen in the band diagram.

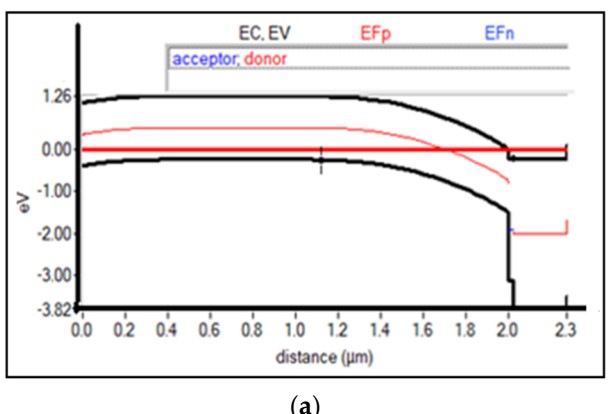

(**a**)

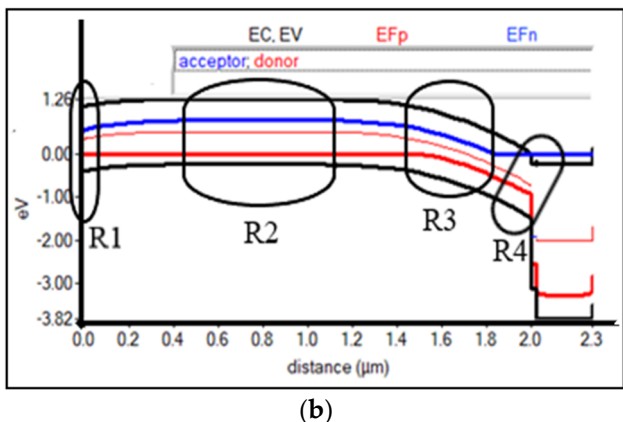

(**b**)

**Figure 8.** The band energy diagram CdTe layer (acceptor). ZnSe layer (buffer) and SnO$_2$ layer window; (**a**) dark current, (**b**) photo current.

Recombination at the back contact (region R1), bulk (quasi–neutral) recombination in the absorber layer (region R2), space charge (region R3) and recombination at the absorber/buffer interface (region R4) are the four regions. The thin absorber layer will maintain the back contact close to the depletion region, resulting in a substantial increase in back contact recombination. Reasonable neutral interface defects for recombination were also included at the mid–gap to accommodate recombination at the CdTe/CdS and ZnSe/SnO$_2$ interfaces [70]. The reflectance of the rear and front contact surfaces was adjusted to 0.1 and 0.9, respectively (Table 3). Photons that traverse the absorber are reflected by this high reflectivity upon return contact, which improves absorption in the absorber.

### 3.4.2. Current Mode

The cross–over and roll–over of the *J–V* curves are the *J–V* characteristics of the Mo/CdTe/ZnSe/SnO$_2$ thin film. The intersection of dark and illuminated *J–V* curves is known as a cross–over. The roll–over phenomenon occurs when the *J–V* curve is meshed and when current levels of greater voltage are present. The dark and photo *J–V* curves are depicted in Figure 9. The ideal layer carrier concentration densities in SCAPS–1D's computation (CdTe $N_A = 2 \times 10^{15}$ cm$^{-3}$, ZnSe $N_D = 1 \times 10^{22}$ cm$^{-3}$ and SnO$_2$ $N_D = 1 \times 10^{22}$ cm$^{-3}$) were used in the simulation. Figure 9 shows the output cell efficiency parameters. The carrier concentration of the absorber layer/buffer layer interface recombination or the absorber/back contact was measured using this advantage [71].

### 3.4.3. Quantum Efficiency

The optimal *QE* for the thin film is shown in Figure 10. The ratio of the number of captured electrons to the number of incident photons on the solar cell is known as the *QE*. The *QE* will be 100% when all the carriers have been gathered and when all the photons have been absorbed by CdTe. Photons ($h\nu \geq E_g$) are absorbed by the absorber layer. Because the absorption layer cannot absorb low–energy photons, high–energy photons are able to contribute to the thermalization process, resulting in a variety of losses, such

as shading losses, spectral mismatch losses, shading losses, incomplete absorption and collection losses, all of which reduce quantum efficiency [67].

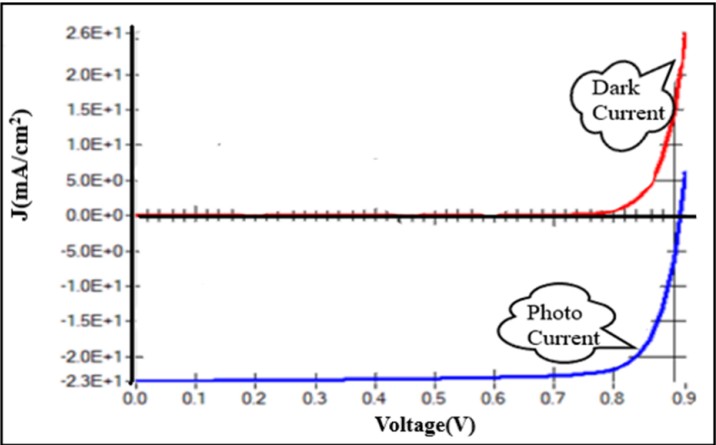

**Figure 9.** Current density with voltage, at $T$ = 300 K, (CdTe $N_A$ = 2 × $10^{15}$ cm$^{-3}$, ZnSe $N_D$ = 1 × $10^{22}$ cm$^{-3}$ and SnO$_2$ $N_D$ = 1 × $10^{22}$ cm$^{-3}$).

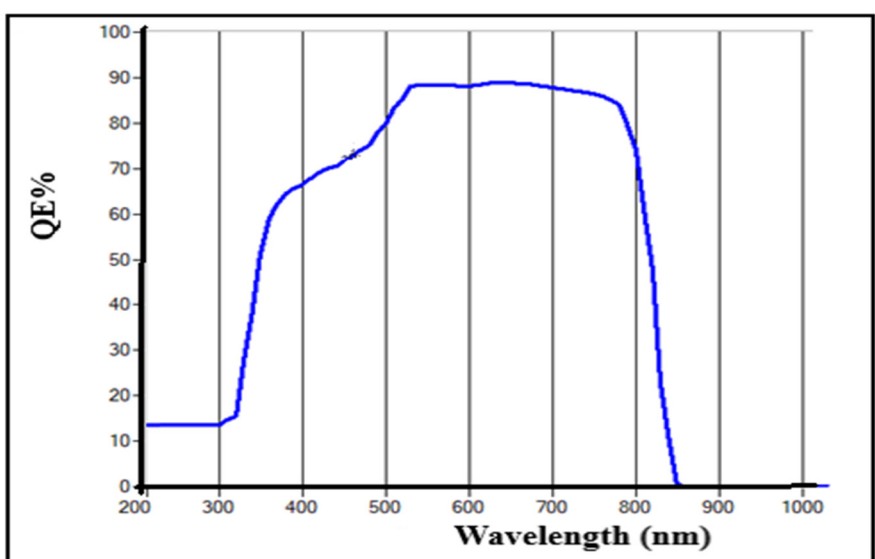

**Figure 10.** Quantum efficiency (*QE*, %) outputs for the thin film (CdTe $N_A$ = 2 × $10^{15}$ cm$^{-3}$, ZnSe $N_D$ = 1 × $10^{22}$ cm$^{-3}$ and SnO$_2$ $N_D$ = 1 × $10^{22}$ cm$^{-3}$), at T = 300 K.

### 3.4.4. Effect of Transparent Conducting Layer (Window Layer)

Both optical and electrical access are provided by transparent conducting oxide (TCO) layers. SnO$_2$ was employed as the TCO in our situation. SnO$_2$ has a bandgap of 3.6 eV, which is sufficient to cover the whole visible wavelength range. SnO$_2$ has a thickness of 250 nm and a donor concentration of 1 × $10^{22}$ cm$^{-3}$, respectively. The window layer contributes to the production of electron–hole pairs slightly.

### 3.5. Comparison between Recent Published Work and Proposed Work

Table 7 shows that the proposed work outperforms recently published studies in terms of the open circuit voltage (V$_{OC}$), shot circuit current (*J$_{SC}$*) and conversion efficiency ($\eta$, %) of the cell construction. The proposed cell structure glass/Mo/CdTe/ZnSe/SnO$_2$ outperforms other cell structures due to high *J$_{SC}$* and V$_{OC}$, which result in higher conversion efficiency. The low *FF* value might be related to defect states in any of the device's layers. If

the proposed cell structure can be effectively manufactured, then this design method will become the superior option.

**Table 7.** Comparison of functional parameters with experimental results.

| | Buffer | $V_{OC}$ (V) | $J_{SC}$ (mA/cm$^2$) | *FF* | $\eta$ (%) | Ref. |
|---|---|---|---|---|---|---|
| CdS | Experimental/CBD | 0.69 | 30.9 | 72 | 15.3 | [72] |
| | Simulated/SCAPS–1D | 0.9113 | 23.4497335 | 81.41 | 17.43 | This work |
| ZnSe | Experimental/CBD | 0.67 | 34.9 | 72.7 | 14.4 | [73] |
| | Simulated/SCAPS–1D | 0.9112 | 23.484037 | 82.38 | 17.42 | This work |
| ZnS | Experimental/CBD | 0.55 | 34.4 | 73 | 13.6 | [73] |
| | Simulated/SCAPS–1D | 0.9121 | 23.260166 | 74.84 | .88 | This work |
| In$_2$S$_3$ | Experimental/ALCVD | 0.27 | 46.8 | 71.5 | 12.9 | [74] |
| | Simulated/SCAPS–1D | 0.9198 | 23.153579 | 66.81 | 14.23 | This work |
| ZnO | Experimental/CBD | 0.835 | 24.1 | 75.46 | 15.19 | [75] |
| | Simulated/SCAPS–1D | 0.9142 | 23.303926 | 76.37 | 16.27 | This work |

## 4. Conclusions

This article employs several buffer layers (CdS, ZnSe, ZnS, In$_2$S$_3$, ZnO) from a numerical simulation standpoint and the outcome indicates that CdS is the best buffer layer. Thus, it can be stated that ZnSe and ZnO are good options as alternate buffer layers for the CdS of CdTe solar cells, which was determined based on the findings from the simulation using SCAPS–1D. Additionally, the material that is used for the CdS buffer layer must be changed to a more appropriate material. Furthermore, numerical simulation analysis has shown that the rise in $N_A$ and $N_D$ results in an increase in solar cell performance. The effect on cell performance was also studied *via* the ZnSe buffer layer. Our analysis also showed that this effect can also be obtained at a value of $\eta$% of 17.42% (with $J_{SC}$ = 23.484037 mA/cm$^2$, $V_{OC}$ = 0.9113 V and *FF* = 81.38) for a 2000 nm thick CdTe absorber layer with $N_A$~2 × 10$^{15}$ cm$^{-3}$, a 25 nm thick ZnSe buffer layer with $N_D$~1 × 10$^{22}$ cm$^{-3}$ and a 250 nm thick SnO$_2$ window layer with $N_D$~1 × 10$^{22}$ cm$^{-3}$. While these results may enable us to create the requested CdTe thin–film solar cell, certain other effective factors have to be investigated in further studies that may influence cell performance.

**Author Contributions:** Data curation, S.H.Z.; formal analysis, S.H.Z. and A.H.Z.; investigation, S.H.Z.; methodology, S.H.Z. and A.H.Z.; project administration, S.H.Z.; resources, S.H.Z.; software, S.H.Z. and N.M.A.; validation, S.H.Z. and A.H.Z.; visualization, S.H.Z.; writing—original draft, S.H.Z.; writing—review & editing, S.H.Z., A.H.Z., N.M.A. and A.F.I.A. All authors have read and agreed to the published version of the manuscript.

**Funding:** This research received no external funding.

**Acknowledgments:** The authors would like to acknowledge the University of Gent, Belgium, for providing the SCAPS simulator. Furthermore, we are grateful to Ajman University for supporting this study.

**Conflicts of Interest:** The authors declare that there are no conflicts of interest in this work.

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
