# Peer review of "Numerical Modelling Analysis for Carrier Concentration Level Optimization of CdTe Heterojunction Thin Film–Based Solar Cell with Different Non–Toxic Metal Chalcogenide Buffer Layers Replacements: Using SCAPS–1D Software"

_crystals, doi:10.3390/cryst11121454_

Round 1
Reviewer 1 Report
The paper title (Numerical Modelling Analysis for Carrier Concentration Level Optimization of CdTe Heterojunction Thin Film–Based Solar Cell with Different Non-Toxic Metal Chalcogenide Buffer Layers Replacements: Using SCAPS-1D Software) presents a numerical simulation to find a suitable non-toxic material for the buffer layer instead of CdS, among various types of buffer layers (ZnSe, ZnO, ZnS, and In2S3), and carrier concentrations.
The manuscript is well written, the methods adequately well described, the results clearly presented and the conclusions supported by the results. This work is mostly good and probably of use to the community. I think the paper has some good merits and deserves publication in crystals journal. I recommend its publication in crystals journal with following minor corrections:
- Reduce the number of figures: Figure 3&Figure 4 in new figure (a,b), Figure 6 & Figure 7 in new figure (a,b), Figure 9 & Figure 10 in new figure (a,b), Figure 12 & Figure 13 in new Figure (a,b).
- Use the same font size for the graphs axis title for example (Font size 20) and same font size and colour for the captions on the graphs (font size 14). Remove the Grid lines from all figures, please.
- Add border to all Figures in (Figure 5,Figure 8, Figure 11)
- Improve the quality of the graphs (r figure resolution 300 dpi) and use one same style and format, font colours, font size, figure border, format legend, arrow colour/ size and shape. Use square comment pane. That’s standard for academic quality paper. Avoid unusable comment shape as in Figure 12.
- Fix Figure 3, Figure 7,Figure 9, and Figure 12 to show the voltage from (0-1.2) V
- Fix Figure 4, Figure 6, Figure 10, and Figure 13 the spectral response to show the wavelength (nm) from (200-1000)
- What does (eta(%) 17.42 mean??) it should Efficiency %. Fix please.
Other than this, I don't have any additional comments.
Author Response
Dear Reviewer,
On the behalf of my co-authors, I would like to thank all of you for careful and thorough reading our manuscript and for the helpful comments and constructive and suggestions, to improve the manuscript. Our response follows your comments sequences. We hope that you find our response satisfactory, and that manuscript in now acceptable for publication in respect Crystals.
Sincerely yours,
On the behalf of co-authors

Reviewer 2 Report
The paper is well prepared and it fits well to the journal scope and topic; some minor comments are pointed out to improve introduction background and figures quality. These can be found below:
- It should be stated in the introduction the relevance of CdTe heterojunction thin-film solar cell within the wide range of different PV technologies. Reviewing reference Quo Vadis Solar Energy Research? Applied Sciences could be referenced for that purpose
- Improve Figure 2 appearance
- Revise explanation in lines 171-72
- Improve Figure 5 appearance and discussion
- Use recommended Table template for Table 6
- Improve Figure 8 appearance
- Improve Figures 10 and 12 quality
Author Response
Dear reviewer,
On the behalf of my co-authors, I would like to thank all of you for careful and thorough reading our manuscript and for the helpful comments and constructive and suggestions, to improve the manuscript. Our response follows your comments sequences. We hope that you find our response satisfactory, and that manuscript in now acceptable for publication in respect Crystals.
Sincerely yours,
On the behalf of co-authors
